# RL-GPT: Integrating Reinforcement Learning and Code-as-policy

**Shaoteng Liu**[1], **Haoqi Yuan**[4], **Minda Hu**[1], **Yanwei Li**[1],
**Yukang Chen,**[1] **Shu Liu,**[2] **Zongqing Lu,**[4,5] **Jiaya Jia**[2,3]

[1] The Chinese University of Hong Kong [2] SmartMore
[3] The Hong Kong University of Science and Technology
[4] School of Computer Science, Peking University
[5] Beijing Academy of Artificial Intelligence
https://sites.google.com/view/rl-gpt/

## Abstract

Large Language Models (LLMs) have demonstrated proficiency in utilizing various tools by coding, yet they face limitations in handling intricate logic and precise control. In embodied tasks, high-level planning is amenable to direct coding, while low-level actions often necessitate task-specific refinement, such as Reinforcement Learning (RL). To seamlessly integrate both modalities, we introduce a two-level hierarchical framework, RL-GPT, comprising a slow agent and a fast agent. The slow agent analyzes actions suitable for coding, while the fast agent executes coding tasks. This decomposition effectively focuses each agent on specific tasks, proving highly efficient within our pipeline. Our approach outperforms traditional RL methods and existing GPT agents, demonstrating superior efficiency. In the Minecraft game, it possibly obtains diamonds within a single day on an RTX3090. Additionally, it achieves good performance on designated MineDojo tasks.

## 1 Introduction

Building agents to master tasks in open-world environments has been a long-standing goal in AI research [1–3]. The emergence of Large Language Models (LLMs) has revitalized this pursuit, leveraging their expansive world knowledge and adept compositional reasoning capabilities [4–6]. LLMs agents showcase proficiency in utilizing computer tools [7, 8], navigating search engines [9, 10], and even operating systems or applications [11, 12]. However, their performance remains constrained in open-world embodied environments [1, 7], such as Minedojo [13]. Despite possessing "world knowledge" akin to a human professor, LLMs fall short when pitted against a child in a video game. The inherent limitation lies in LLMs' adeptness at absorbing information but their inability to practice skills within an environment. Proficiency in activities such as playing a video game demands extensive practice, a facet not easily addressed by in-context learning, which exhibits a relatively low upper bound [7, 4, 6]. Consequently, existing LLMs necessitate human intervention to define low-level skills or tools.

Reinforcement Learning (RL), proven as an effective method for learning from interaction, holds promise in facilitating LLMs to "practise". One line of works grounds LLMs for open-world control through RL fine-tuning [14–19]. Nevertheless, this approach necessitates a substantial volume of domain-specific data, expert demonstrations, and access to LLMs' parameters, rendering it slow and resource-intensive in most scenarios. Given the modest learning efficiency, the majority of methods continue to operate within the realm of "word games" such as tone adjustment rather than tackling intricate embodied tasks.

38th Conference on Neural Information Processing Systems (NeurIPS 2024).

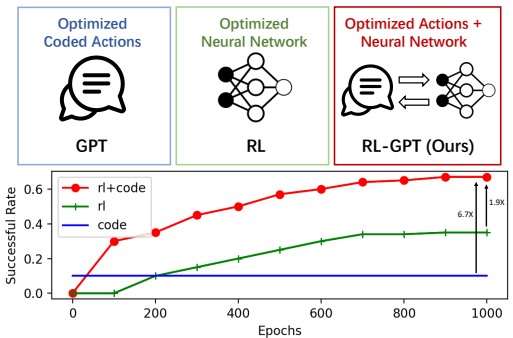

Figure 1: An overview of RL-GPT. After the optimization in an environment, LLMs agents obtain optimized coded actions, RL achieves an optimized neural network, and our RL-GPT gets both optimized coded actions and neural networks. Our framework integrates the coding parts and the learning parts.

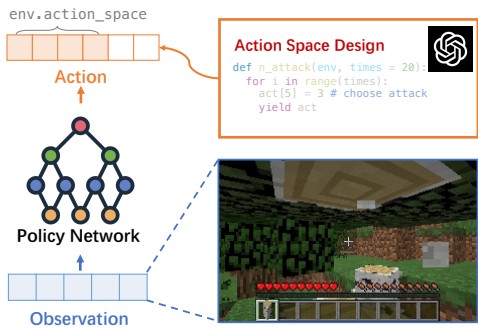

Figure 2: To learn a subtask, the LLM can generate environment configurations (task, observation, reward, and action space) to instantiate RL. In particular, by reasoning about the agent behavior to solve the subtask, the LLM generates code to provide higher-level actions in addition to the original environment actions, improving the sample efficiency for RL.

Addressing this challenge, we propose to integrate LLMs and RL in a novel approach: *Empower LLMs agents to use an RL training pipeline as a tool.* To this end, we introduce RL-GPT, a framework designed to enhance LLMs with trainable modules for learning interaction tasks within an environment. As shown in Fig. 3, RL-GPT comprises an agent pipeline featuring multiple LLMs, wherein the neural network is conceptualized as a tool designed for training the RL pipeline. Illustrated in Fig. 1, unlike conventional approaches where LLMs agents and RL optimize coded actions and networks separately, RL-GPT unifies this optimization process. The line chart in Fig. 1 illustrates that RL-GPT outperforms alternative approaches by seamlessly integrating both knowledge iteration and skill practice.

We further point out that the pivotal issue in using RL is to decide: *Which actions should be learned with RL?* To tackle this, RL-GPT is meticulously designed to assign different actions to RL and Code-as-policy [20], respectively. Our agent pipeline entails two fundamental steps. Firstly, LLMs should determine "which actions" to code, involving task decomposition into distinct sub-actions and deciding which actions can be effectively coded. Actions falling outside this realm will be learned through RL. Secondly, LLMs are tasked with writing accurate codes for the "coded actions" and test them in the environment.

We employ a two-level hierarchical framework to realize the two steps, as depicted in Fig. 3. Allocating these steps to two independent agents proves highly effective, as it narrows down the scope of each LLM's task. Coded actions with explicit starting conditions are executed sequentially, while other coded actions are integrated into the RL action space. This strategic insertion into the action space empowers LLMs to make pivotal decisions during the learning process. Illustrated in Fig. 2, this integration enhances the efficiency of learning tasks, exemplified by our ability to more effectively learn how to break a tree.

For intricate tasks such as the ObtainDiamond task in the Minecraft game, devising a strategy with a single neural network proves challenging due to limited computing resources. In response, we incorporate a task planner to facilitate task decomposition. Our RL-GPT framework demonstrates remarkable efficiency in tackling complex embodied tasks. Specifically, within the MineDojo environment, it attains good performance on the majority of selected tasks and adeptly locates diamonds within a single day, utilizing only an RTX3090 GPU. Our contributions are summarized as follows:

- Introduction of an LLMs agent utilizing an RL training pipeline as a tool.

- Development of a two-level hierarchical framework capable of determining which actions in a task should be learned.

- Pioneering work as the first to incorporate high-level GPT-coded actions into the RL action space.

## 2 Related Works

### 2.1 Agents in Minecraft

Minecraft, a widely popular open-world sandbox game, stands as a formidable benchmark for constructing efficient and generalized agents. Previous endeavors resort to hierarchical reinforcement learning, often relying on human demonstrations to facilitate the training of low-level policies [21–23]. Efforts such as MineAgent [13], Steve-1 [24], and VPT [25] leverage large-scale pre-training via YouTube videos to enhance policy training efficiency. However, MineAgent and Steve-1 are limited to completing only a few short-term tasks, and VPT requires billions of RL steps for long-horizon tasks. DreamerV3 [26] utilizes a world model to expedite exploration but still demands a substantial number of interactions to acquire diamonds. These existing works either necessitate extensive expert datasets for training or exhibit low sample efficiency when addressing long-horizon tasks.

An alternative research direction employs Large Language Models (LLMs) for task decomposition and high-level planning to offload RL's training burden using LLMs' prior knowledge. Certain works [27] leverage few-shot prompting with Codex [28] to generate executable policies. DEPS [29] and GITM [30] investigate the use of LLMs as high-level planners in the Minecraft context. VOY-AGER [1] and Jarvis-1 [31] explore LLMs for high-level planning, code generation, and lifelong exploration. Other studies [32, 33] delve into grounding smaller language models for control with domain-specific finetuning. Nevertheless, these methods often rely on manually designed controllers or code interfaces, sidestepping the challenge of learning low-level policies.

Plan4MC [34] integrates LLM-based planning and RL-based policy learning but requires defining and pre-training all the policies with manually specified environments. Our RL-GPT extends LLMs' ability in low-level control by equipping it with RL, achieving automatic and efficient task learning.

### 2.2 LLMs Agents

Several works leverage LLMs to generate subgoals for robot planning [35, 36]. Inner Monologue [37] incorporates environmental feedback into robot planning with LLMs. Code-as-Policies [20] and ProgPrompt [38] directly utilize LLMs to formulate executable robot policies. VIMA [39] and PaLM-E [2] involve fine-tuning pre-trained LLMs to support multimodal prompts. Chameleon [4] effectively executes sub-task decomposition and generates sequential programs. ReAct [6] utilizes chain-of-thought prompting to generate task-specific actions. AutoGPT [7] automates NLP tasks by integrating reasoning and acting loops. DERA [40] introduces dialogues between GPT-4 [41] agents. Generative Agents [42] simulate human behaviors by memorizing experiences. Creative Agent [43] achieves creative building generation in Minecraft. Our paper equips LLMs with RL to explore environments.

### 2.3 Integrating LLMs and RL

Since LLMs and RL possess complementary abilities in providing prior knowledge and exploring unknown information, it is promising to integrate them for efficient task learning. Prior works include using decision trees, finite state machines, DSL programs, and symbolic programs as policies [44–50].

Most work studies improve RL with the domain knowledge in LLMs. SayCan [35] and Plan4MC [34] decompose and plan subtasks with LLMs, thereby RL can learn easier subtasks to solve the whole task. Recent works [51–54] studies generating reward functions with LLMs to improve the sample efficiency for RL. Some works [55–59] used LLMs to train RL agents. Other works [14, 15, 60, 61, 16, 17, 62] finetune LLMs with RL to acquire the lacked ability of LLMs in low-level control. However, these approaches usually require a lot of samples and can harm the LLMs' abilities in other tasks. Our study is the first to overcome the inabilities of LLMs in low-level control by equipping them with RL as a tool. The acquired knowledge is stored in context, thereby continually improving the LLMs skills and maintaining its capability.

## 3 Methods

Our framework employs a decision-making process to determine whether an action should be executed using code or RL. The RL-GPT incorporates three distinct components, each contributing to its

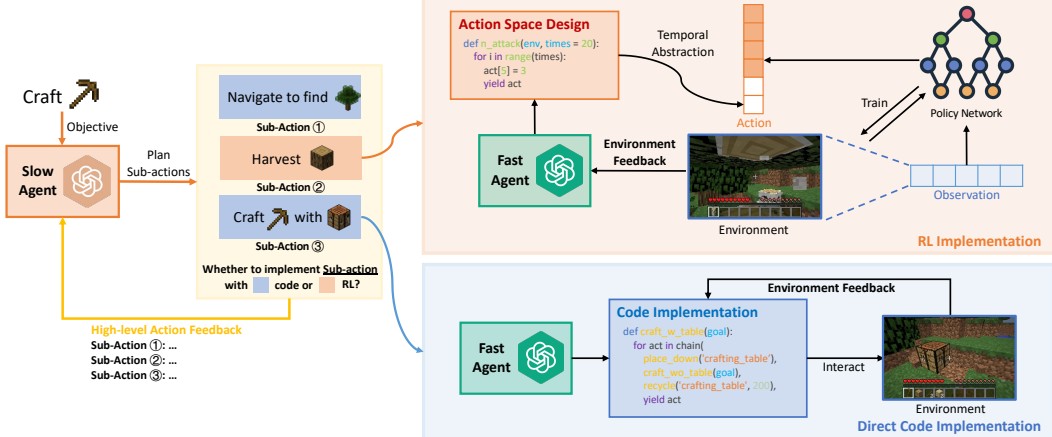

Figure 3: Overview of RL-GPT. The overall framework consists of a slow agent (orange) and a fast agent (green). The slow agent decomposes the task and determines "which actions" to learn. The slow agent will improve the decision based on the high-level action feedbacks. The fast agent writes code and RL configurations. The fast agent debugs the written code based on the environment feedback ("Direct Code Implementation"). Correct codes will be inserted into the action space as high-level actions ("RL Implementation").

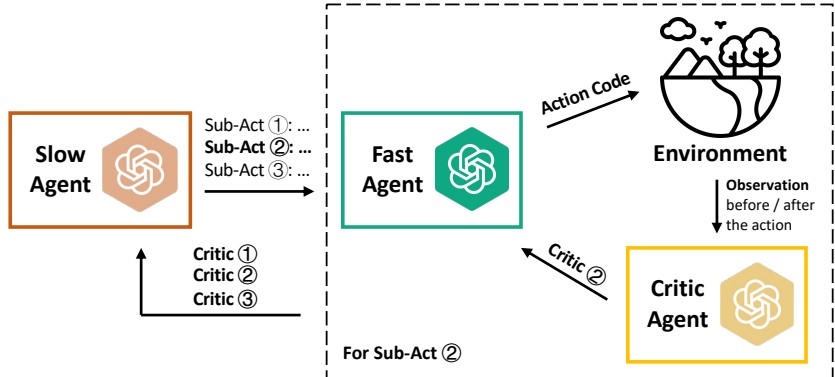

Figure 4: The two-loop iteration. We design a method to optimize both slow agent and fast agent with a critic agent.

innovative design: (1) a slow agent tasked with decomposing a given task into several sub-actions and determining which actions can be directly coded, (2) a fast agent responsible for writing code and instantiating RL configuration, and (3) an iteration mechanism that facilitates an iterative process refining both the slow agent and the fast agent. This iterative process enhances the overall efficacy of the RL-GPT across successive iterations. For complex long-horizon tasks requiring multiple neural networks, we employ a GPT-4 as a planner to initially decompose the task.

As discussed in concurrent works [63, 64], segregating high-level planning and low-level actions into distinct agents has proven to be beneficial. The dual-agent system effectively narrows down the specific task of each agent, enabling optimization for specific targets. Moreover, Liang et al. highlighted the Degeneration-of-Thought (DoT) problem, where an LLM becomes overly confident in its responses and lacks the ability for self-correction through self-reflection. Empirical evidence indicates that agents with different roles and perspectives can foster divergent thinking, mitigating the DoT problem. External feedback from other agents guides the LLM, making it less susceptible to DoT and promoting accurate reasoning.

## 3.1 RL Interface

As previously mentioned, we view the RL training pipeline as a tool accessible to LLMs agents, akin to other tools with callable interfaces. Summarizing the interfaces of an RL training pipeline, we

identify the following components: *1) Learning task; 2) Environment reset; 3) Observation space; 4) Action space; 5) Reward function.* Specifically, our focus lies on studying interfaces 1) and 4) to demonstrate the potential for decomposing RL and Code-as-policy.

In the case of the action space interface, we enable LLMs to design high-level actions and integrate them into the action space. A dedicated token is allocated for this purpose, allowing the neural network to learn when to utilize this action based on observations.

## 3.2 Slow Agent: Action Planning

We consider a task $T$ that can feasibly be learned using our current computing resources (e.g., lab-level GPUs). We employ a GPT-4 [41] as a slow agent $A_S$. $A_S$ is tasked with decomposing $T$ into sub-actions $\alpha_i$, where $i \in \{0, ..., n\}$, determining if each $\alpha_i$ in $T$ can be directly addressed through code implementation. This approach optimally allocates computational resources to address more challenging sub-tasks using Reinforcement Learning techniques. Importantly, $A_S$ is not required to perform any low-level coding tasks; it solely provides high-level textual instructions including the detailed description and context for sub-actions $\alpha_i$. These instructions are then transmitted to the fast agent $A_F$ for further processing. The iterative process of the slow agent involves systematically probing the limits of coding capabilities.

For instance, in Fig. 3, consider the specific action of crafting a wooden pickaxe. Although $A_S$ is aware that players need to harvest a log, writing code for this task with a high success rate can be challenging. The limitation arises from the insufficient information available through APIs for $A_S$ to accurately locate and navigate to a tree. To overcome this hurdle, an RL implementation becomes necessary. RL aids $A_S$ in completing tasks by processing complex visual information and interacting with the environment through trial and error. In contrast, straightforward actions like crafting something with a table can be directly coded and executed.

It is crucial to instruct $A_S$ to identify sub-actions that are too challenging for rule-based code implementation. As shown in Table 6, the prompt for $A_S$ incorporates role description **{role_description}**, the given task $T$, reference documents, environment knowledge **{minecraft_knowledge}**, planning heuristics **{planning_tips}**, and programming examples **{programs}**. To align $A_S$ with our goals, we include the heuristic in the **{planning_tips}**. This heuristic encourages $A_S$ to further break down an action when coding proves challenging. This incremental segmentation aids $A_S$ in discerning what aspects can be coded. Further details are available in Appendix A.

## 3.3 Fast Agent: Code-as-Policy and RL

The fast agent $A_F$ is also implemented using GPT-4. The primary task is to translate the instructions from the slow agent $A_S$ into Python codes for the sub-actions $\alpha_i$. $A_F$ undergoes a debug iteration where it runs the generated sub-action code and endeavors to self-correct through feedback from the environment. Sub-actions that can be addressed completely with code implementation are directly executed, as depicted in the blue segments of Fig. 3. For challenging sub-actions lacking clear starting conditions, the code is integrated into the RL implementation using the temporal abstraction technique [65, 66], as illustrated in Fig. 2. This involves inserting the high-level action into the RL action space, akin to the orange segments in Fig. 3. $A_F$ iteratively corrects itself based on the feedback received from the environment.

## 3.4 Two-loop Iteration

In Fig. 4, we have devised a two-loop iteration to optimize the proposed two agents, namely the fast agent $A_F$ and the slow agent $A_S$. To facilitate it, a critic agent $C$ is introduced, which could be implemented using GPT-3.5 or GPT-4.

The optimization for the fast agent, as shown in Fig. 4, aligns with established methods for code-as-policy agents. Here, the fast agent receives a sub-action, environment documents $D_{env}$ (observation and action space), and examples $E_{code}$ as input, generating Python code. It then iteratively refines the code based on environmental feedback. The objective is to produce error-free Python-coded sub-actions that align with the targets set by the slow agent. Feedback, which includes execution errors and critiques from $C$, plays a crucial role in this process. $C$ evaluates the coded action's success by considering observations before and after the action's execution.

Within Fig. 4, the iteration of the slow agent $A_S$ encompasses the aforementioned fast agent $A_F$ iteration as a step. In each step of $A_S$, $A_F$ must complete an iteration loop. Given a task $T$, $D_{env}$, and $E_{code}$, $A_S$ decomposes $T$ into sub-actions $\alpha_i$ and refines itself based on $C$'s outputs. The critic's output includes: (1) whether the action is successful, (2) why the action is successful or not, (3) how to improve the action. Specifically, it receives a sequence of outputs $Critic_i$ from $C$ about each $\alpha_i$ to assess the effectiveness of action planning. If certain actions cannot be coded by the fast agent, the slow agent adjusts the action planning accordingly.

### 3.5   Task Planner

Our primary pipeline is tailored for tasks that can be learned using a neural network within limited computational resources. However, for intricate tasks such as ObtainDiamond, where it is more effective to train multiple neural networks like DEPS [29] and Plan4MC [34], we introduce a task planner reminiscent of DEPS, implemented using GPT-4. This task planner iteratively reasons what needs to be learned and organizes sub-tasks for our RL-GPT to accomplish.

## 4   Experiments

### 4.1   Environment

**MineDojo**  MineDojo [13] stands out as a pioneering framework developed within the renowned Minecraft game, tailored specifically for research involving embodied agents. This innovative framework comprises a simulation suite featuring thousands of tasks, blending both open-ended challenges and those prompted by language. To validate the effectiveness of our approach, we selected certain long-horizon tasks from MineDojo, mirroring the strategy employed in Plan4MC [34]. These tasks include harvesting and crafting activities. For instance, Crafting one wooden pickaxe requires the agent to harvest a log, craft planks, craft sticks, craft tables, and craft the pickaxe with the table. Similarly, tasks like milking a cow involve the construction of a bucket, approaching the cow, and using the bucket to obtain milk.

**ObtainDiamond Challenge**  It represents a classic challenge for RL methods. The task of obtaining a diamond demands the agent to complete the comprehensive process of harvesting a diamond from the beginning. This constitutes a long-horizon task, involving actions such as harvesting logs, harvesting stones, crafting items, digging to find iron, smelting iron, locating a diamond, and so on.

### 4.2   Implementation Details

**LLM Prompt**  We choose GPT-4 as our LLMs API. For the slow agents and fast agents, we design special templates, responding formats, and examples. We design some special prompts such as "assume you are an experienced RL researcher that is designing the RL training job for Minecraft". Details can be found in the Appendix A. In addition, we encourage the slow agent to explore more strategies because the RL task requires more exploring. We encourage the slow agent to further decompose the action into sub-actions which may be easier to code.

**PPO Details**  The training and evaluation are the same as Mineagent or other RL pipelines as discussed in Appendix C. The difference is that our RL action space contains high-level coded actions generated by LLMs. Our method doesn't depend on any video pretraining. It can work with only environment interaction. Similar to MineAgent [13], we employ Proximal Policy Optimization (PPO) [67] as the RL baseline. This approach alternates between sampling data through interactions with the environment and optimizing a "surrogate" objective function using stochastic gradient ascent. PPO is constrained to a limited set of skills. When applying PPO with sparse rewards, specific tasks such as "milk a cow" and "shear a sheep" present challenges due to the small size of the target object relative to the scene, and the low probability of random encounters. To address this, we introduce basic dense rewards to enhance learning efficacy in these tasks. It includes the CLIP [68] Reward, which encourages the agent to exhibit behaviors that align with the prompt [13]. Additionally, we incorporate a Distance Reward that provides dense reward signals to reach the target items [34]. It costs 3K steps for harvesting a log referring to Table. 16. For the diamond task, the evaluation ends when the diamond is found or the agent is dead. It will cost around 20K steps. The frame rate for the game is 30 fps. Further details can be found in the appendix C.

Table 1: Comparison of several tasks selected from the Minedojo benchmark. Our RL-GPT achieves the highest success rate on all tasks. All values in our tables refer to the actual successful rate.

| TASK | | | | | | | | | | |
|---|---|---|---|---|---|---|---|---|---|---|
| MINEAGENT | 0.00 | 0.00 | 0.00 | 0.00 | 0.00 | 0.00 | -- | -- | -- | -- |
| MINEAGENT (AUTOCRAFT) | 0.00 | 0.03 | 0.00 | 0.00 | 0.00 | 0.46 | 0.50 | 0.33 | 0.35 | 0.00 |
| PLAN4MC | 0.30 | 0.30 | 0.53 | 0.37 | 0.17 | 0.83 | 0.53 | 0.43 | 0.33 | 0.17 |
| RL-GPT | **0.65** | **0.65** | **0.67** | **0.67** | **0.64** | **0.85** | **0.56** | **0.46** | **0.38** | **0.32** |

Table 2: Main results in the challenging ObtainDiamond task in Minecraft. Existing strong baselines in ObtainDiamond either require expert data (VPT, DEPS), hand-crafted policies (DEPS-Oracle) for subtasks, or take huge number of environment steps to train (DreamerV3, VPT). Our method can automatically decompose and learn subtasks with only a little human prior, achieving ObtainDiamond with great sample efficiency.

| METHOD | TYPE | SAMPLES | SUCCESS |
|---|---|---|---|
| DREAMERV3 | RL | 100M | 2% |
| VPT | IL+RL | 16.8B | 20% |
| DEPS-BC | IL+LLM | -- | 0.6% |
| DEPS-ORACLE | LLM | -- | 60% |
| PLAN4MC | RL+LLM | 7M | 0% |
| RL-GPT | RL+LLM | 3M | 8% |

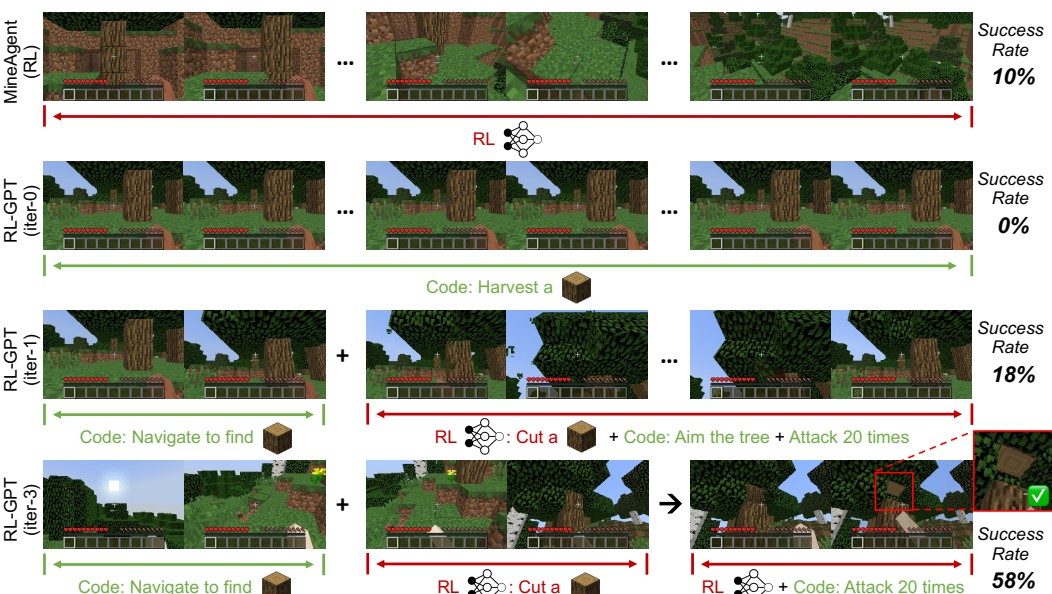

Figure 5: Demonstrations of how different agents learn to harvest a log. While both RL agent and LLM agent learn a single type of solution (RL or code-as-policy), our RL-GPT can reasonably decompose the task and correct how to learn each sub-action through the slow iteration process. RL-GPT decomposes the task into "find a tree" and "cut a log", solving the former with code generation and the latter with RL. After a few iterations, it learns to provide RL with a necessary high-level action (attack 20 times) and completes the task with a high success rate. Best viewed by zooming in.

## 4.3 Main Results

**MineDojo Benchmark**    Table 1 presents a comparative analysis between our RL-GPT and several baselines on selected MineDojo tasks. Notably, RL-GPT achieves the highest success rate among all

Table 3: Ablation on the RL and Code-as-policy components and the iteration mechanism. RL-GPT outperforms its Pure RL and Pure Code counterparts. Given more iterations, RL-GPT gets better results.

| Method | 🪵 | ⛏ | 🪣 | 🛏 |
|---|---|---|---|---|
| Pure RL | 0.00 | 0.00 | 0.00 | 0.00 |
| Pure Code | 0.13 | 0.02 | 0.00 | 0.00 |
| Ours (Zero-shot) | 0.26 | 0.53 | 0.79 | 0.32 |
| Ours (Iter-2 w/o SP) | 0.26 | 0.53 | 0.79 | 0.30 |
| Ours (Iter-2) | 0.56 | 0.67 | 0.88 | 0.30 |
| Ours (Iter-3) | **0.65** | **0.67** | **0.93** | **0.32** |

Table 4: Ablation on the agent structure.

| Structure | 🪵 | ⛏ |
|---|---|---|
| One Agent | 0.34 | 0.42 |
| Slow + Fast | 0.52 | 0.56 |
| Slow + Fast + Critic | **0.65** | **0.67** |

Table 5: Ablation on the RL interfaces.

| Interface | Success ↑ | Dead Loop ↓ |
|---|---|---|
| Reward | 0.418 | ≈0.6 |
| Action | **0.585** | **≈0.3** |

baselines. All baselines underwent training with 10 million samples, and the checkpoint with the highest success rate was chosen for testing.

MineAgent, as proposed in [13], combines PPO with CLIP Reward. However, naive PPO encounters difficulties in learning long-horizon tasks, such as crafting a bucket and obtaining milk from a cow, resulting in an almost 0% success rate for MineAgent across all tasks. Another baseline, MineAgent with autocraft, as suggested in Plan4MC [34], incorporates crafting actions manually coded by humans. This alternative baseline achieves a 46% success rate on the milking task, demonstrating the importance of code-as-policy. Our approach demonstrates superiority in coding actions beyond crafting, enabling us to achieve higher overall performance compared to these baselines.

Plan4MC [34] breaks down the problem into two essential components: acquiring fundamental skills and planning based on these skills. While some skills are acquired through Reinforcement Learning (RL), Plan4MC outperforms MineAgent due to its reliance on an oracle task decomposition from the GPT planner. However, it cannot modify the action space of an RL training pipeline or flexibly decompose sub-actions. It is restricted to only three types of human-designed coded actions. Consequently, our method holds a distinct advantage in this context.

In tasks involving ✏ and 🪵, the agent is tasked with crafting a stick from scratch, necessitating the harvesting of a log. Our RL-GPT adeptly codes three actions for this: *1) Navigate to find a tree; 2) Attack 20 times; 3) Craft items.* Notably, Action 2) can be seamlessly inserted into the action space. In contrast, Plan4MC is limited to coding craft actions only. This key distinction contributes to our method achieving higher scores in these tasks.

To arrive at the optimal code planning solution, RL-GPT undergoes a minimum of three iterations. As illustrated in Fig. 5, in the initial iteration, RL-GPT attempts to code every action involved in harvesting a log, yielding a 0% success rate. After the first iteration, it decides to code navigation, aiming at the tree, and attacking 20 times. However, aiming at the tree proves too challenging for LLMs. As mentioned before, the agent will be instructed to further decompose the actions and give up difficult actions. By the third iteration, the agent correctly converges to the optimal solution—coding navigation and attacking, while leaving the rest to RL, resulting in higher performance.

In tasks involving crafting a wooden pickaxe ⛏ and crafting a bed 🛏, in addition to the previously mentioned actions, the agent needs to utilize the crafting table. While Plan4MC must learn this process, our method can directly code actions to place the crafting table on the ground, use it, and recycle it. Code-as-policy contributes to our method achieving a higher success rate in these tasks.

In tasks involving crafting a furnace 🔲 and a stone pickaxe ⛏, in addition to the previously mentioned actions, the agent is further required to harvest stones. Plan4MC needs to learn an RL network to acquire the skill of attacking stones. RL-GPT proposes two potential solutions for coding additional actions. First, it can code to continuously attack a stone and insert this action into the action space. Second, since LLMs understand that stones are underground, the agent might choose to dig deep for several levels to obtain stones instead of navigating on the ground to find stones.

In crafting a milk bucket 🪣 and crafting wool 🐑, the primary challenge is crafting a bucket or shears. Since both RL-GPT and Plan4MC can code actions to craft without a crafting table, their performance is comparable. Similarly, obtaining beef 🥩 and obtaining mutton 🍖 needs navigating.

**ObtainDiamond Challenge**    As shown in Tab. 2, we compare our method with existing competitive methods on the challenging ObtainDiamond task.

DreamerV3 [26] leverages a world model to accelerate exploration but still requires a significant number of interactions. Despite the considerable expense of over 100 million samples for learning, it only achieves a 2% success rate on the Diamond task from scratch.

VPT [25] employs large-scale pre-training using YouTube videos to improve policy training efficiency. This strong baseline is trained on 80 GPUs for 6 days, achieving a 20% success rate in obtaining a diamond and a 2.5% success rate in crafting a diamond pickaxe.

DEPS [29] suggests generating training data using a combination of GPT and human handcrafted code for planning and imitation learning. It attains a 0.6% success rate on this task. Moreover, an oracle version, which directly executes human-written codes, achieves a 60% success rate.

Plan4MC [34] primarily focuses on crafting the stone pickaxe. Even with the inclusion of all human-designed actions from DEPS, it requires more than 7 million samples for training.

Our RL-GPT attains an over 8% success rate in the ObtainDiamond challenge by generating Python code and training a PPO RL neural network. Despite requiring some human-written code examples, our approach uses considerably fewer than DEPS. The final coded actions involve navigating on the ground, crafting items, digging to a specific level, and exploring the underground horizontally.

## 4.4   Ablation Study

**Framework Structure**    In Tab. 4, we analyze the impact of the framework structure in RL-GPT, specifically examining different task assignments for various agents. Assigning all tasks to a single agent results in confusion due to the multitude of requirements, leading to a mere 34% success rate in crafting a table. Additionally, comparing the 3rd and 4th rows emphasizes the crucial role of a critic agent in our pipeline. Properly assigning tasks to the fast, slow, and critic agents can improve the performance to 65%. Slow agent faces difficulty in independently judging the suitability of actions based solely on environmental feedback and observation. Incorporating a critic agent facilitates more informed decision-making, especially when dealing with complex, context-dependent information.

**Two-loop Iteration**    In Tab. 3, we ablate the importance of our two-loop iteration. Our iteration is to balance RL and code-as-policy to explore the bound of GPT's coding ability. We can see that pure RL and pure code-as-policy only achieve a low success rate on these chosen tasks. Our method can improve the results although there is no iteration (zero-shot). In these three iterations, it shows that the successful rate increases. It proves that the two-loop iteration is a reasonable optimization choice. Qualitative results can be found in Fig. 5. Besides, we also compare the results with and without special prompts (SP) to encourage the LLMs to further decompose actions when facing coding difficulty. It shows that suitable prompts are also essential for optimization.

**RL Interface**    Recent works [51, 52] explore the use of LLMs for RL reward design, presenting an alternative approach to combining RL and code-as-policy. With slight modifications, our fast agent can also generate code to design the reward function. However, as previously analyzed, reconstructing the action space proves more efficient than designing the reward function, assuming LLMs understand the necessary actions. Tab. 5 compares our method with the reward design approach. Our method achieves a higher average success rate and lower dead loop ratio on our selected MineDojo tasks.

## 5   Conclusion

In conclusion, we propose RL-GPT, a novel approach that integrates Large Language Models (LLMs) and Reinforcement Learning (RL) to empower LLMs agents in practicing tasks within complex, embodied environments. Our two-level hierarchical framework divides the task into high-level coding and low-level RL-based actions, leveraging the strengths of both approaches. RL-GPT exhibits superior efficiency compared to traditional RL methods and existing GPT agents, achieving remarkable performance in the challenging Minecraft environment.

**Acknowlegdement**  This work was supported in part by the Research Grants Council under the Areas of Excellence scheme grant AoE/E-601/22-R.

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

# Appendix

## A  Agent Prompt Details

Prompt details of fast and slow agent including **{role_description}**, **{planning_tips}**,**{act_info}**, and **{obs_info}** are listed in Table 8,9,10. We provide several human-written code blocks following DEPS [29]. We list some codes in Table 11. These human-provided codes are for crafting tasks since it is difficult for GPT-4 to write correct crafting actions. Our agent will not directly use these crafting codes as API. It will writing code by itself based on these contexts. We also provide the critic agent's prompt details in Table 13. We have provided a comprehensive overview of helpful and unhelpful low-level skills written by LLMs in Table 15 and Table 14. These examples showcase the range of skills generated by LLMs, including those beneficial for task completion and others that are less effective. The useful low-level skills identified will be integrated into the action space, as illustrated in Fig. 2 of our paper. It's important to note that unhelpful low-level skills typically fall into two categories: (1) Wrong in details: Examples include actions like attacking a tree a limited number of times or attempting to "use" an item ineffectively for a crafting task. (2) Too difficult actions: While actions such as adjusting sight to the tree before an attack may be reasonable, they can be overly complex to code and execute efficiently.

---

**{role_description}**

It is difficult to code all actions in this game. We only want to code as many sub-actions as possible. The task of you is to tell me which sub-actions can be coded by you with Python.

At each round of conversation, I will give you
Task: $T$
Context: ...
Critique: The results of the generated codes in the last round

Here are some actions coded by humans:
**{programs}**

You should then respond to me with
Explain (if applicable): Why these actions can be coded by python? Are there any actions difficult to code?
Actions can be coded: List all actions that can be coded by you.

Important Tips:
**{planning_tips}**

You should only respond in the format as described below:

Explain: ...
Actions can be coded:
1) Action1: ...
2) Action2: ...
3) ...

Table 6: Slow Agent's prompt: Decompose a task into sub-actions.

---

**{role_description}**

Here are some basic actions coded by humans:
**{programs_template}**

Please inherit the class CodeAgent. You are only required to overwrite the function main_function.

Here are some reference examples written by me:
**{programs_example}**

Here are the attributes of the obs that can be used:
**{obs_info}**

Here are the guidelines of the act variable:
**{act_info}**

At each round of conversation, I will give you
Task: ...
Context: ...
Code from the last round: ...
Execution error: ...
Critique: ...

You should then respond to me with
Explain (if applicable): Can the code complete the given action? What does the chat log and execution error imply?

You should only respond in the format as described below:
**{code_format}**

Table 7: Fast Agent's prompt: Write Python codes.

**{role_description}:**
You are playing the game Minecraft. Assume you are a Python programmer. You want to write python code to complete some parts of this game.

**{planning_tips}:**
1) If it is unsuccessful to code one action in the last round, it means the action is too difficult for coding.
2) If one action in the last round is too difficult to code, try to further subdivide the action. For example, if "attacking the tree 20 times" is difficult, try "simply attacking 20 times".
3) Please refer to the additional knowledge about Minecraft. It is very useful.

Table 8: Slow Agent's prompt details

**{role_description}:**
We want to write python code to complete some actions in Minecraft. You are a helpful assistant that helps to write the code for the given action tasks.

**{act_info}:**
We design a compound action space. At each step the agent chooses one movement action (forward, backward, camera actions, etc.) and one optional functional action (attack, use, craft, etc.). Some functional actions such as craft take one argument, while others like attack does not take any argument. This compound action space can be modelled in an autoregressive manner.

Technically, our action space is a multi-discrete space containing eight dimensions:
$>>>$ *env.action_space*
MultiDiscrete([3, 3, 4, 25, 25, 8, 244, 36])

Index 0; Forward and backward; 0: noop, 1: forward, 2: back
Index 1; Move left and right; 0: noop, 1: move left, 2: move right
Index 2; Jump, sneak, and sprint; 0: noop, 1: jump, 2: sneak, 3:sprint
Index 3; Camera delta pitch; 0: -180 degree, 24: 180 degree
Index 4; Camera delta yaw; 0: -180 degree, 24: 180 degree
Index 5; Functional actions; 0: noop, 1: use, 2: drop, 3: attack, 4: craft, 5: equip, 6: place, 7: destroy
Index 6; Argument for "craft"; All possible items to be crafted
Index 7; Argument for "equip", "place", and "destroy"; Inventory slot indice

Table 9: Fast Agent's prompt details

**obs["rgb"]:**
RGB frames provide an egocentric view of the running Minecraft client that is the same as human players see.
*Data type:* numpy.uint8
*Shape:* (3, H, W), height and width are specified by argument image_size

**obs["inventory"]["name"]:**
Names of inventory items in natural language, such as "obsidian" and "cooked beef".
*Data type:* str
*Shape:* (36,)

We also provide voxels observation (3x3x3 surrounding blocks around the agent). This type of observation is similar to how human players perceive their surrounding blocks. It includes names and properties of blocks.

**obs["voxels"]["block_name"]:**
Names of surrounding blocks in natural language, such as "dirt", "air", and "water".
*Data type:* str
*Shape:* (3, 3, 3)

**obs["location_stats"]["pos"]:**
The xyz position of the agent.
*Data type:* numpy.float32
*Shape:* (3,)

**obs["location_stats"]["yaw"] and obs["location_stats"]["pitch"]:**
Yaw and pitch of the agent.
*Data type:* numpy.float32
*Shape:* (1,)

**obs["location_stats"]["biome_id"]:**
Biome ID of the terrain the agent currently occupies.
*Data type:* numpy.int64
*Shape:* (1,)

Lidar observations are grouped under obs["rays"]. It includes three parts: information about traced entities, properties of traced blocks, and directions of lidar rays themselves.

**obs["rays"]["entity_name"]:**
Names of traced entities.
*Data type:* str
*Shape:* (num_rays,)

**obs["rays"]["entity_distance"]:**
Distances to traced entities.
*Data type:* numpy.float32
*Shape:* (num_rays,)

Properties of traced blocks include blocks' names and distances from the agent.

**obs["rays"]["block_name"]:**
Names of traced blocks in natural language in the fan-shaped area ahead of the agent, such as "dirt", "air", and "water".
*Data type:* str
*Shape:* (num_rays,)

**obs["rays"]["block_distance"]:**
Distances to traced blocks in the fan-shaped area ahead of the agent.
*Data type:* numpy.float32
*Shape:* (num_rays,)

Table 10: Observation information **{obs_info}** of Fast Agent

```python
# look to a specific direction
def look_to(self, deg = 0):
    #accquire info
    obs, reward, done, info = self.accquire_info()
    obs_ = self.env.obs_
    while obs["location_stats"]["pitch"] < deg:
        act = self.env.action_space.no_op()
        act[3] = 13
        act[5] = 3
        yield act
        obs, reward, done, info = self.accquire_info()

    while obs["location_stats"]["pitch"] > deg:
        act = self.env.action_space.no_op()
        act[3] = 11
        act[5] = 3
        yield act
        obs, reward, done, info = self.accquire_info()

# place the item in the hands
def place(self, goal):
    slot = self.index_slot(goal)
    if slot == -1:
        return False

    act = self.env.action_space.no_op()
    act[2] = 1
    act[5] = 6
    act[7] = slot
    yield act

# place the table in the hands and use it
def place_down(self, goal):
    if self.index_slot(goal) == -1:
        return None

    for act in chain(
        self.look_to(deg=83),
        self.attack(2),
        self.place(goal),
    ):
        yield act

# recycle the table after using it
def recycle(self, goal, times = 20):
    for i in range(times):
        act = self.env.action_space.no_op()
        act[5] = 3
        obs, reward, done, info = self.env.step(act)
        if any([item['name'] == goal for item in info['inventory']]):
            break

    yield self.env.action_space.no_op()
    for act in chain( self.look_to(0), self.take_forward(3), ):
        yield act
```

Table 11: Human-written code examples for crafting actions. (Continued in Table 12)

```python
# directly craft something without a crafting table
def craft_wo_table(self, goal):
    act = self.env.action_space.no_op()
    act[5] = 4
    act[6] = self.craft_smelt_items.index(goal)
    print(goal, self.craft_smelt_items.index(goal))
    yield act

# craft something with a crafting table
def craft_w_table(self, goal):
    #print('here', self.index_slot('crafting_table'))
    if self.index_slot('crafting_table') == -1:
        return None

    for act in chain(
        self.place_down('crafting_table'),
        self.craft_wo_table(goal),
        self.recycle('crafting_table', 200),
    ):
        print(f"goal: act")
        yield act

# smelt something with a furnace
def smelt_w_furnace(self, goal):
    #print('Here', self.index_slot('furnace'))
    if self.index_slot('furnace') == -1:
        return None

    for act in chain(
        self.place_down('furnace'),
        self.craft_wo_table(goal),
        self.recycle('furnace', 200),
    ):
        yield act

# directly smelt something without a furnace
def smelt_wo_furnace(self, goal):
    for act in self.craft_wo_table(goal):
        yield act
```

Table 12: Human-written code examples for crafting actions.

We want to write python code to complete some actions in Minecraft.
You are an assistant that assesses whether the coded actions are effective. Can the code complete the target action?
You need to analyze why the code is successful or not. Please give detailed reasoning and critique.

I will give you the following information:

To code the action: The action we want to code.
Context for the action: The context of the coded action.
Code: The code written by gpt to complete the action.

Observations before running the coded action:
Inventory Name: Names of inventory items in natural languages, such as "obsidian" and "cooked beef".
Inventory Quantity
Blocks in lidar rays: Names of traced blocks.
Entities in lidar rays: Names of traced entities.
Around blocks: Names of surrounding blocks in natural language, such as "dirt", "air", and "water".

Observations after running the coded action:
Inventory Name
Inventory Quantity
Blocks in lidar rays
Entities in lidar rays
Around blocks

You should only respond in JSON format as described below:
{
"reasoning": "reasoning",
"success": boolean,
"critique": "critique",
}

Table 13: Critic Agent's prompt details

```
# unuseful skills
# continuously attacking 10 times (not enough to break a log)
for act in chain(
      self.attack(10),
):
      yield act

# "use" instead of "craft"
act[5] = 3
act[6] = self.craft_smelt_items.index(goal)

# look at the tree (difficult to succeed)
while not self.target_in_sight(obs, 'wood', max_dis=5):
      act[3] = 13
      act[5] = 3
      yield act

# look at the tree and attack (the tree may not be in the front)
for act in chain(
      self.look_to_front,
      self.attack,
):
      yield act
```

Table 14: Unuseful low-level skills designed by the LLMs

```
# useful skills

# continuously attacking 20 times
def attack(self, times = 20):
    for i in range(times):
        act[5] = 3
        yield act

for act in chain(
    self.attack(20),
):
    yield act

# craft an item
act[5] = 4
act[6] = self.craft_smelt_items.index(goal)

# look to the front
while obs["location_stats"]["pitch"] < 50:
    act[3] = 13
    act[5] = 3
    yield act
while obs["location_stats"]["pitch"] > 50:
    act[3] = 11
    act[5] = 3
    yield act

# move forward
for i in range(10):
    act[0] = 1
    yield act

# navigate to find a cow
if random.randint(0, 20) == 0:
    act[4] = 1
if random.randint(0, 20) == 0:
    act[0] = 1
for act in chain(
    self.look_to_front,
    self.forward,
    self.attack,
):
    yield act

# place the crafting table
slot = self.index_slot('crafting_table')
act[2] = 1
act[5] = 6
act[7] = slot
yield act

# mine deep to a depth
if self.env.obs["location_stats"]["pos"][1] > depth:
    if self.env.obs["location_stats"]["pitch"] < 80:
        act[3] = 13
        yield act
    else:
        act[5] = 3
        yield act
```

Table 15: Useful low-level skills designed by the LLMs.

# B Algorithms

---

**Algorithm 1** RL-GPT's Two-loop Iteration

---

**Input:** Task $T$, Slow agent $A_S$, Fast agent $A_F$, Critic agent $C$, Prompt for slow agent $P_S$, Prompt for fast agent $P_F$. Critic$_i$ = None
**repeat**
    $\alpha_0, ..., \alpha_n = A_S(T, P_S)$
    **for** $i = 0$ **to** $n$ **do**
        **repeat**
            Code = $A_F(\alpha_i, P_F, \text{Critic}_i)$
            act_space = rl_config(Code)
            Obs$_i$ = rl_training(act_space)
            Critic$_i$ = $C_F$(rl_config, code, Obs$_i$)
        **until** no bug
    **end for**
    $P_S = P_S + \text{Critic}_0 + ... + \text{Critic}_n$
**until** $T$ is complete

---

Actions deemed unsuitable for coding are not included in $\alpha_0, ..., \alpha_n$, ensuring they are learned naturally through RL neural networks during training.

# C Details in PPO

**CLIP reward.** The reward incentivizes the agent to generate behaviors aligned with the task prompt. 31 task prompts are selected from the entire set of MineDojo programmatic tasks as negative samples. Utilizing the pre-trained MineCLIP model [13], we calculate the similarities between the features extracted from the past 16 frames and the prompts. The probability is then computed, indicating the likelihood that the frames exhibit the highest similarity to the given task prompt: $p = [\text{softmax}\left(S\left(f_v, f_l\right), \{S\left(f_v, f_{l-}\right)\}_{l-}\right)]_0$, where $f_v, f_l$ are video features and prompt features, $l$ is the task prompt, and $l^-$ are negative prompts. The CLIP reward is:

$$r_{\text{CLIP}} = \max\left\{p - \frac{1}{32}, 0\right\}. \tag{1}$$

**Distance reward.** The distance reward offers dense reward signals for reaching target items. In combat tasks, the agent receives a distance reward when the current distance is closer than the minimum distance observed in history:

$$r_{distance} = \max\left\{\min_{t' < t} d_{t'} - d_t, 0\right\}. \tag{2}$$

For mining tasks, where the agent needs to remain close to the block for several time steps, we adapt the distance reward to promote maintaining a small distance:

$$r_{distance} = \begin{cases} d_{t-1} - d_t, & 1.5 \le d_t \le +\infty \\ 2, & d_t < 1.5 \\ -2, & d_t = +\infty, \end{cases} \tag{3}$$

where $d_t$ is the distance between the agent and the target item at time step $t$, detected through lidar rays in the simulator.

# D Other Environments

The powerful zero-shot capability of GPT serves as a guarantee of its generalization ability. It also represents an advantage of RL-GPT over pure RL methods. Our method mainly works in complex environments with both high-level planning and low-level controlling, like the real world. In robotic tasks requiring long-horizon planning and motor execution, RL-GPT is a promising framework to

Table 16: Settings for MineDojo tasks in our paper.

| Task Icon | Target Name | Initial Tools | Biome | Max Steps |
|---|---|---|---|---|
| | stick | -- | plains | 3000 |
| | crafting_table_nearby | -- | plains | 3000 |
| | wooden_pickaxe | -- | forest | 3000 |
| | furnace_nearby | *10 | hills | 5000 |
| | stone_pickaxe | | forest_hills | 10000 |
| | milk_bucket | , *3 | plains | 3000 |
| | wool | , *2 | plains | 3000 |
| | beef | | plains | 3000 |
| | mutton | | plains | 3000 |
| | bed | , | plains | 10000 |

Table 17: OpenAI tokens it takes to "speed up" RL.

| | ITER-1 | ITER-2 | ITER-3 |
|---|---|---|---|
| TOKENS | 10K | 15K | 16K |

decompose tasks and learn subtasks with different solutions autonomously. Subtasks requiring simple locomotion skills, such as navigation and reaching, might be easily acquired with code-as-policy via motion planning. Nevertheless, there are other complex skills, such as object manipulation, which necessitate using RL tools to address. In Fig. 6, we present a qualitative demonstration in the Furniture environment [69]. The motion planning action effectively aids in hole-finding tasks during table assembly, highlighting the practical utility of our approach in complex assembly scenarios. Fig. 7 illustrates the RL training process in the Kitchen environment [70]. The vertical axis represents the success rate, and the horizontal axis represents the number of training steps. Inserting coded motion planning into the action space accelerates learning. Our method learns faster compared to the baseline. In Fig. 8, we present a qualitative demonstration of the Furniture environment [69]. The motion planning action effectively aids in hole-finding tasks during table and chair assembly. The baseline struggles to find the correct location at the same training step. Modifications are needed for different domains, such as adjusting the task descriptions in the prompts. The powerful zero-shot capability of GPT should ensure generalization ability.

# E  Other Details

We count the average tokens on different tasks for the first 3 iterations in Tab. 17. Works like Voyager only consider high-level planning, using human-coded skill libraries to bypass the need for low-level control. Our method considers both high-level planning and low-level actions, directly facing the Minedojo action space. To acquire low-level policies autonomously, RL is necessary. We visualize these in Tab. 18. We show the comparison on the "harvest a log" task in Tab. 19. The performance of Claude is similar to GPT-4. Vicuna-13b has lower performance due to its poor coding ability. VLMs can function as more effective critic agents in our framework. While LLMs can only indicate whether the agent succeeded with its coded actions, VLMs can explain why it failed in the environment. As shown in Fig. 9, GPT-4V provides more detailed feedback across different environments. For example, in Minecraft, it can identify that the agent keeps attacking the ground instead of finding the cow. In the driving simulation environment, it can note that the vehicle is gradually drifting off the road. This feedback can be used by both our fast and slow agents for self-improvement.

Table 18: High-level comparison among different methods.

| METHOD | LONG-HORIZON TASK | LOW-LEVEL CONTROL | SAMPLE-EFFICIENCY | SELF-IMPROVEMENT |
|---|---|---|---|---|
| MINEAGENT | ✗ | ✓ | ✗ | ✗ |
| VPT | ✓ | ✓ | ✗ | ✗ |
| DEPS | ✓ | ✗ | ✓ | ✗ |
| VOYAGER | ✓ | ✗ | ✓ | ✓ |
| RL-GPT | ✓ | ✓ | ✓ | ✓ |

Table 19: Comparison among different LLMs.

| LLMs | SUCCESS RATE | DEAD LOOP |
|---|---|---|
| VICUNA-13B | 0.36 | 0.8 |
| CLAUDE | 0.64 | 0.3 |
| GPT-4 | 0.65 | 0.3 |

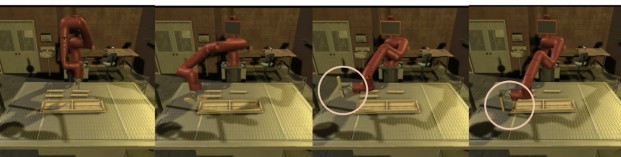

*Baseline: Insert in the wrong position*

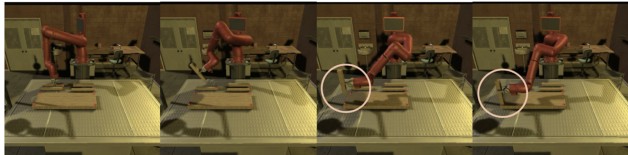

*Ours: Motion planning (as an action) to the hole*

Figure 6: Qualitative results on Furniture. Using motion planning as an action can help the robot arm find the hole more efficiently.

# F   Limitations

This framework relies on the LLMs being able to initialize the environment and code reasonable actions. As a result, the upper bound of our method is limited by LLMs' ability. LLMs may exhibit limited out-of-sample generalization capabilities, meaning they may struggle to generalize on unseen data. This implies that in real-world applications, models may exhibit unpredictable or unstable behavior under unknown circumstances, posing challenges to system security. This is particularly critical in domains requiring high reliability and stability, such as autonomous driving or medical devices. Besides, due to the expense of GPT, experiments are completed with fewer random seeds.

# G   Broader Impact

The integration of RL-GPT in real-world interactions presents exciting opportunities but also raises security concerns. While RL-GPT facilitates seamless collaboration between agents and humans, enabling efficient task execution and adaptive responses, its reliance on large-scale language models introduces vulnerabilities. These vulnerabilities include susceptibility to adversarial attacks, where malicious inputs can manipulate the model's behavior, potentially leading to unintended actions or compromised system integrity. Privacy and security protection are crucial issues when using LLMs for natural language processing. LLMs may handle vast amounts of personal data and sensitive information during training and deployment, necessitating stringent privacy protection measures to prevent data breaches or misuse, thereby safeguarding user privacy and security.

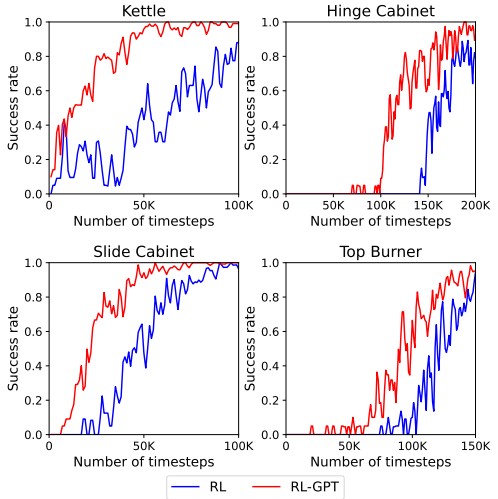

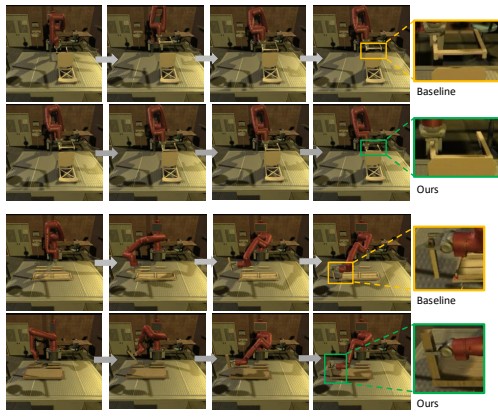

Figure 7: The training curves in the Kitchen environment. Integrating coded motion planning and RL accelerates learning. RL-GPT learns faster compared to the RL baseline.

Figure 8: Qualitative demonstrations in the Furniture environment show that our motion planning action effectively aids in furniture assembly, whereas baselines struggle to find the correct location.

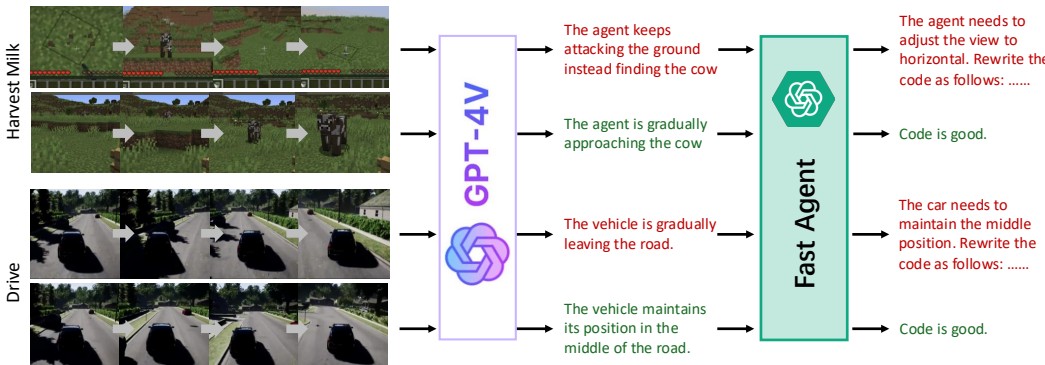

Figure 9: VLMs can offer more precise critiques based on vision feedback.

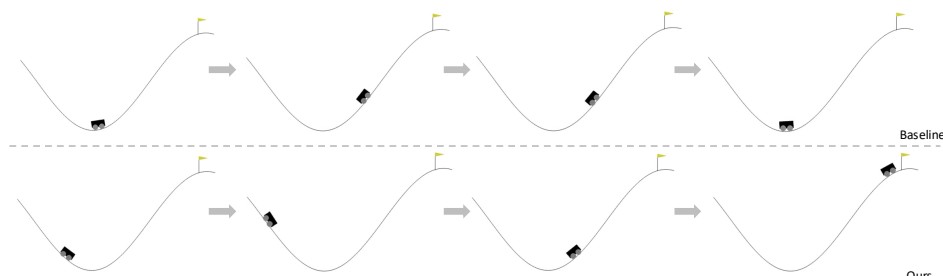

Figure 10: MuJoCo example. GPT-4 can code an action to reverse the car and then move it forward.

