# OpenReview forum: "RL-GPT: Integrating Reinforcement Learning and Code-as-policy"
_NeurIPS.cc/2024/Conference — NeurIPS 2024 oral_

### Official Review · Reviewer_ep8Y · 2024-06-30

**Soundness:** 2
**Presentation:** 2
**Contribution:** 3
**Rating:** 5
**Confidence:** 3

**Summary:**

The authors present a framework to give LLMs the ability to code and train RL agents as a tool for completing tasks. They perform experiments in MineDojo.

**Strengths:**

**Experiments:** Barring the standard deviation issue mentioned in teh weaknesses, the results are a significant improvement over Plan4MC.

**Idea:** the method of using RL as a tool for crafting subfunctions to complete tasks via LLMs is interesting, intuitive, and simple.

**Weaknesses:**

**Minor Issues:**

- some relevant related works using LLMs to train RL agents: [https://rlvlmf2024.github.io/,](https://rlvlmf2024.github.io/) https://mihdalal.github.io/planseqlearn/, https://clvrai.github.io/boss/, https://clvrai.github.io/sprint/, https://instructionaugmentation.github.io/, https://gimme1dollar.github.io/lift/,

**Clarity:**

- The term “Code-as-Policy” is introduced without definition or citations in the introduction. Is this a common term? Is this referring to the [Code as Policies](https://code-as-policies.github.io/) paper? This will confuse readers who haven’t seen this term before.
- The related works section doesn’t do a great job differentiating this work against prior work. For example section 2.2 doesn’t mention the current work in relation to prior work at all, same with the first paragraph of 2.3.
- The method section isn’t really clear, it’s missing details, references to appendix sections, or references to figures clarifying things. Examples:
    - L133: If the focus is not on the reward function then how is the reward obtained?
    - L136: “A dedicated token is allocated for [integrating high-level actions]…” how is this trained or used?
    - L181: What are the inputs and outputs of $C$? image observatoins or other details from the environment state?
    - Overall, there’s not enough examples or details here to understand the method and how things are learned/trained without having to thoroughly dig through the appendix. For space reasons obviously not all details can be here, but I think this methods section can be rewritten to be much better.
- Experiments:
    - L230 “It costs 3k steps for harvesting a log” what does this mean? This is the minimum number of timesteps required? or is this the empirical number the RL agent that the authors trained requires?
    - L232: please link to a specific section in the appendix.

**Experiments:**

- There are no standard deviations on any numbers, or information about # seeds in the main paper. How many seeds did you run?
- Task-directed Voyager seems like a very direct comparison that uses LLMs for everything instead of RL. Why is this not compared?

**Questions:**

See weaknesses

**Limitations:**

Yes, but i think it's typical to include this in the main paper instead of the appendix.

---

> ### Author Rebuttal · Authors · 2024-08-07
>
> Dear Reviewer ep8Y,
>
> Despite the negative score, we really appreciate your detailed review. We address your questions below.
>
> **Q1. Related works.**
>
> **A1.** Thanks for mentioning these papers. We will add some of them to the related work part in the revision.
> - RL-VLM-F introduces visual feedback to the reward function design.
> - Plan-Seq-Learn combines motion planning and RL skills to solve long-horizon tasks, using LLM as the high-level planner.
> - BOSS chains short-term skills to construct long-term skills with LLM supervision.
> - SPRINT uses LLMs to relabel instructions.
> - DIAL uses foundation models for dataset augmentation.
> - LiFT uses foundation models for task instruction proposal and reward design.
>
> The motivation, technical novelty, and experimental results of our work distinctively set it apart from these works:
> - Our **motivation** is to equip LLMs with RL capabilities to explore environments and autonomously decide which actions to learn using RL.
> - Our **technical novelty** involves inserting high-level actions coded by LLMs into the RL action space.
> - Our **main results** demonstrate that our agent can successfully determine which actions to learn and complete long-horizon targets in Minecraft.
>
> We will modify Sections 2.2 and 2.3 following your suggestions.
>
> **Q2. Code-as-Policy.**
>
> **A2.** Sorry for the confusion. Yes, the definition of "Code-as-Policy" in this paper refers to LLMs writing code that is then executed in the environment to control the agent. We will reference this paper to clarify this concept.
>
> **Q3. Writing issues.**
>
> **A3.** Thanks for these valuable suggestions!
> - L133: It means our agent will not focus on reward design. Instead, default reward functions, such as sparse rewards and distance rewards, are used for agent training in the environment.
> - L136: It means that one of the output dimensions of the RL network will represent the coded high-level action. This high-level action will be executed when the network selects that option.
> - L181: You can find the outputs in lines **185 to 187**. The inputs are simply observations from the environment, as shown in **Tab. 10**.
>
> We will clarify our method based on your suggestions in future versions.
>
> **Q4. Experiment issues.**
>
> **A4.** Thanks for these valuable suggestions!
> - L230: For these tasks, the maximum exploration steps are capped at 3K. We will compare the success rate for these tasks. More details can be found in **Tab. 16**.
> - L232: It is linked to **Section C**.
> - Seeds: For quantitative ablation studies on harvest tasks, we used **3 seeds**. Due to the large number of tasks, the rest were tested on 1 seed. This approach aligns with practices in the field. Given the low simulation speed of the Minecraft game, other significant works, such as VPT [1], also report outcomes based on a single RL training seed.
>
> [1] Video PreTraining (VPT): Learning to Act by Watching Unlabeled Online Videos, 2022
>
> Works like Voyager focus solely on **high-level** planning, using human-coded skill libraries to bypass the need for **low-level control**. Our method integrates both high-level planning and learning low-level skills, directly addressing the challenge of playing the Minecraft game using keyboard and mouse actions. RL is necessary to acquire low-level policies autonomously.
>
> - We are the first agent that utilizes an RL training pipeline as a tool and decides which actions to be learned by RL.
> - We are the first to incorporate GPT-coded actions into the RL action space, enhancing the sample efficiency for RL.
> - The critic agent is similar to Voyager, but Voyager doesn't have a two-level hierarchical framework since it only contains high-level planning.
>
> Here is a system-level comparison table:
>
> |   Method   | long-horizon task | low-level control | sample-efficiency | self-improvement |
> |  ----  | ----  | ----  | ----  | ----  |
> | MineAgent  | &#10007; |	&#10004; | &#10007; | &#10007; |
> | VPT  | &#10004; | &#10004;  | &#10007; | &#10007; |
> | DEPS  | &#10004; | &#10007; | &#10004; | &#10007; |
> | Voyager  | &#10004; | &#10007; | &#10004; | &#10004; |
> | RL-GPT  | &#10004; | &#10004; | &#10004; | &#10004; |

---

> > ### Comment · Reviewer_ep8Y · 2024-08-12
> >
> > Thanks for the response!. Overall, due to NeurIPS' rebuttal format, I cannot determine how the writing issues will be addressed (unless authors provide direct quotes for each part).
> > However, I have re-read the paper again, with the proposed changes in mind, and I also saw some areas where I simply missed the author's text in answering some of my questions.
> >
> > Regarding experiments, I fundamentally disagree that experiments in simulation should only be run with 1 seed. VPT does use just one seed, but just because one paper uses one seed doesn't mean that should be the standard everywhere.
> >
> > I'm raising my score but keeping it at a borderline simply due to this standard deviation issue as I believe the paper itself has merit.

---

> > > ### Author Response · Authors · 2024-08-12
> > >
> > > Thank you for your thoughtful feedback and for raising your score! We appreciate your insights and will carefully revise the writing issues you highlighted. We also take your point about using multiple seeds seriously and will incorporate experiments with multiple seeds in the revision.

---

### Official Review · Reviewer_efny · 2024-07-10

**Soundness:** 3
**Presentation:** 4
**Contribution:** 3
**Rating:** 8
**Confidence:** 4

**Summary:**

This paper introduces RL-GPT, a hierarchical framework that uses LLMs to first break down complex embodied tasks into sub-actions that are suitable for coding or learnable through RL, and then write codes or RL configurations to execute the actions. The authors evaluate the framework on MineDojo benchmark and the challenging ObtainDiamond task, showcasing better performance and efficiency compared to existing baselines.

**Strengths:**

1. The integration of RL training pipeline is novel and generally applicable to other domains.
2. Strong empirical results on Minecraft tasks. Detailed ablation studies on key design choices.
3. The paper is clearly written and well presented.

**Weaknesses:**

The proposed framework is only evaluated on the MineCraft games, which the GPTs have extensive knowledge about due to the massive relevant contents on the internet. It's unclear if the framework could be easily extended to novel domains like simple MuJoCo simulated environments, new games, or more real-world tasks, e.g. navigation or household tasks with real robots.

**Questions:**

1. Have you tried using any smaller / open-source LLMs instead of GPT-4? How does the performance change?
2. When the fast agent fails to code up a sub-action, how does the slow agent decide if it should further break down the action into steps, or use an RL training pipeline?
3. Does the framework keep an archive of solutions (high level plans + codes / RL agents)?

**Limitations:**

Yes.

---

> ### Author Rebuttal · Authors · 2024-08-07
>
> Dear Reviewer efny,
>
> Thank you for appreciating our work with valuable suggestions. We address your questions below.
>
> **Q1. Generalization to other environments.**
>
> **A1.** We acknowledged this concern and addressed it to some extent in Appendix Section D. It is difficult to find real-world environments like Minecraft, which require **both high-level planning and low-level control**. We applied our methods to some robotic tasks that demand both long-horizon planning and precise motor execution. Some results are shown in **Fig. 7 and Fig. 8 in the attached pdf**.
>
> - Kitchen Environment Training: **Fig.7** illustrates the RL training process in the Kitchen environment[1]. The vertical axis represents the success rate, and the horizontal axis represents the number of training steps. Inserting coded motion planning into the action space accelerates learning. Our method learns faster compared to the baseline.
> - Furniture Environment Demonstration: In **Fig.8**, we present a qualitative demonstration of the Furniture environment[2]. The motion planning action effectively aids in hole-finding tasks during table and chair assembly. The baseline struggles to find the correct location at the same training step.
>
> [1] Relay Policy Learning: Solving Long-Horizon Tasks via Imitation and Reinforcement Learning, 2019
>
> [2] Furniture Assembly Environment for Long-Horizon Complex Manipulation Tasks, 2021
>
> Additionally, VLMs can be integrated into our critic agent, enabling future expansion to vision-based environments like driving, as shown in **Fig. 9** of the attached pdf. Simple MuJoCo simulated environments may not be ideal for our method due to their **focus on low-level controls** and lack of high-level planning or decision-making. However, our method performs well in some cases, as illustrated in **Fig. 10**. For instance, GPT-4 can code an action to reverse the car and then move it forward based on the topography.
> Modifications are needed for different domains, such as adjusting the task descriptions in the prompts. The powerful zero-shot capability of GPT should ensure generalization ability.
>
> **Q2. Smaller / open-source LLMs.**
>
> **A2.** Thanks for this good question! Here is the comparison on the ''harvest a log'' task. The performance of Claude is similar to GPT-4. Vicuna-13b has lower performance due to its poor coding ability. Mixtral-8x7B works much better than Vicuna-13b. Open-sourced methods are continuously improving, making them promising for future agent development.
>
> |   LLMs   | Success Rate | Dead loop |
> |  ----  | ----  | ----  |
> | Vicuna-13b | 0.36 | 0.8 |
> | Mixtral-8x7B | 0.55 | 0.5 |
> | Claude  | 0.64 | 0.3 |
> | GPT-4  | 0.65 | 0.3 |
>
> **Q3. When the fast agent fails to code up a sub-action.**
>
> **A3.**
> Thanks for this good question! When the fast agent is unable to code an action, it implies that at least part of the action requires RL training. The slow agent will decompose this action, which is a process of gradually analyzing which specific sub-action needs RL. For the steps that can be coded, the code is written. If decomposition fails to resolve the issue after a certain number of iterations, the task will be handed over to RL.
>
> **Q4. Keeping an archive of solutions.**
>
> **A4.** Yes, both successful coded actions and well-trained RL networks will be preserved during agent optimization. They will be executed as skills with specific names, similar to Voyager's vector database.

---

> > ### Comment · Reviewer_efny · 2024-08-13
> >
> > Thanks for the response. I have increased my score given the additional results.

---

> > > ### Author Response · Authors · 2024-08-13
> > >
> > > Thank you for recognizing our work and raising the score. We sincerely appreciate your valuable suggestions and will ensure they are incorporated in the revision.

---

### Official Review · Reviewer_m13r · 2024-07-13

**Soundness:** 3
**Presentation:** 3
**Contribution:** 3
**Rating:** 5
**Confidence:** 3

**Summary:**

This work is a variation of the code as policies, which utilizes LLMs to write code robotic policies in code snippets.  This work examines minecraft, proposing that certain tasks can be composed into two sets: those solvable using LLM generated code and those best left to be solved using a standard RL agent.  They utilize 2 LLM prompting styles.  The first is for an LLM agent which decomposes tasks and determines which ones can be learned as code or using RL.  The 2nd actually implements the code and inserts it into the action space for use by the RL agent.  A critic LLM determines if the action was successful and how to improve it.  This is used for iterative improvement of the code generating LLM.

**Strengths:**

This deals with one of the most important problems in utilizing LLMs and code as policies, the fact that some actions are just not well suited for code and should be learned using standard RL.

Ablations including removing the critic are shown.

The improvements in Minecraft appear to be substantial.

Overall, the paper presents a good improvement in an area of interest to many RL researchers. I generally support acceptance of this work if the limitations/broader impact are discussed.

**Weaknesses:**

Alot of manual design is needed for each specific environment.

A large amount of calls to an LLM API are used and this method becomes very expensive quickly.

I would like to see how VLMs could be leveraged in other environments.

The work does not explore envs outside of Minecraft.  I would like to see how applications of this work could extend to other common RL envs.

**Questions:**

Can the authors discuss the broader impact of this method in envs outside of Minecraft and how it can apply to other types of environments in robotics?

**Limitations:**

Limitations are clearly the expense of this work, could the authors discuss a bit more about this?

---

> ### Author Rebuttal · Authors · 2024-08-07
>
> Dear Reviewer m13r,
>
> Thank you for appreciating our work with valuable suggestions. We address your questions below.
>
> **Q1. Manual design is needed for each specific environment.**
>
> **A1.** Yes, we acknowledge that some manual design is necessary. However, compared to existing agents like Voyager, which require humans to write low-level action code, our **RL training reduces manual design by 90%**. As LLMs' zero-shot capabilities continue to advance, the need for manual design will further decrease.
>
> **Q2. Expensive API call.**
>
> **A2.** Thanks for this good question! We agree that calling API has a cost.
> Here are the statistics. We count the average tokens on different tasks for the first 3 iterations.
>
> |      | iter-1 | iter-2 | iter-3 |
> |  ----  | ----  | ----  | ----  |
> | tokens  | 10K | 15K  | 16K  |
>
> GPT-4-32k will cost $0.12 per 1,000 tokens. The newly released **GPT-4o Mini** will be more cost-effective.
>
> Here is the comparison on the ''harvest a log'' task. Vicuna-13b has lower performance due to its poor coding ability. Mixtral-8x7B works much better than Vicuna-13b. Open-sourced methods are continuously improving, making them promising for future agent development.
>
> |   LLMs   | Success Rate | Dead loop |
> |  ----  | ----  | ----  |
> | Vicuna-13b | 0.36 | 0.8 |
> | Mixtral-8x7B | 0.55 | 0.5 |
> | Claude  | 0.64 | 0.3 |
> | GPT-4  | 0.65 | 0.3 |
>
> **Q3. How VLMs could be leveraged in other environments?**
>
> **A3.** Thanks for this good question! VLMs can function as **more effective critic agents** in our framework. While LLMs can only indicate whether the agent succeeded with its coded actions, VLMs can explain why it failed in the environment. As shown in **Fig. 9 of the attached pdf**, GPT-4V provides **more detailed feedback** across different environments. For example, in Minecraft, it can identify that the agent keeps attacking the ground instead of finding the cow. In the driving simulation environment, it can note that the vehicle is gradually drifting off the road. This feedback can be used by both our fast and slow agents for self-improvement.
>
> **Q4. Generalization to other environments.**
>
> **A4.** We acknowledged this concern and addressed it to some extent in Appendix Section D. It is difficult to find real-world environments like Minecraft, which require **both high-level planning and low-level control**. We applied our methods to some robotic tasks that demand both long-horizon planning and precise motor execution. Some results are shown in **Fig. 7 and Fig. 8 in the attached pdf**.
>
> - Kitchen Environment Training: **Fig.7** illustrates the RL training process in the Kitchen environment[1]. The vertical axis represents the success rate, and the horizontal axis represents the number of training steps. Inserting coded motion planning into the action space accelerates learning. Our method learns faster compared to the baseline.
> - Furniture Environment Demonstration: In **Fig.8**, we present a qualitative demonstration of the Furniture environment[2]. The motion planning action effectively aids in hole-finding tasks during table and chair assembly. The baseline struggles to find the correct location at the same training step.
>
> [1] Relay Policy Learning: Solving Long-Horizon Tasks via Imitation and Reinforcement Learning, 2019
>
> [2] Furniture Assembly Environment for Long-Horizon Complex Manipulation Tasks, 2021
>
> Modifications are needed for different domains, such as adjusting the task descriptions in the prompts. The powerful zero-shot capability of GPT should ensure generalization ability.

---

### Official Review · Reviewer_LWBp · 2024-07-21

**Soundness:** 3
**Presentation:** 3
**Contribution:** 4
**Rating:** 7
**Confidence:** 4

**Summary:**

The paper introduces RL-GPT, a novel framework that integrates Large Language Models (LLMs) with Reinforcement Learning (RL) to enhance the performance of LLM-based agents in complex, embodied environments. The primary goal is to address the limitations of LLMs in executing intricate logic and precise control, especially in open-world tasks like those found in the Minecraft game.

The RL-GPT framework employs a two-level hierarchical approach, consisting of a slow agent and a fast agent. The slow agent is responsible for decomposing tasks and determining which actions can be coded directly, while the fast agent generates the corresponding code and integrates RL configurations. This division of labor allows for efficient handling of both high-level planning and low-level execution.

Key contributions of the paper include:
1. Two-level hierarchical framework to determine which actions should be learned by RL and which ones can be coded (e.g. in a python).
2. The introduction of a mechanism where high-level actions coded by LLMs are appended to the RL agent’s action space, instead of only relying on low level actions.
3. Empirical validation showing that RL-GPT outperforms traditional RL methods and existing LLM agents in the Minecraft environment, particularly excelling in the ObtainDiamond task and other MineDojo tasks.

In experiments, RL-GPT demonstrated superior performance, achieving state-of-the-art (SOTA) results in several tasks, including rapidly obtaining diamonds and excelling in long-horizon tasks with limited computational resources.

**Strengths:**

**Originality**:
- The paper proposes a novel integration of RL and LLMs, leveraging the strengths of both to improve task efficiency in open-world environments.
- The fast agent's method of combining code-as-policy with RL by inserting high-level coded actions into the RL action space is a creative and original solution that sets RL-GPT apart from other frameworks.

**Quality**:
- Claims are well supported by extensive experimental results, comparing with previous SOTA methods in MineDojo


**Clarity**:
- The paper is generally well-written and structured, with detailed explanations of the framework components and their interactions.
- Figures and tables effectively illustrate the performance improvements and workflow of RL-GPT.


**Significance**:
- There is often a question about the what is the right abstraction level for LLM agents to operate on: high level APIs/code-as-policies, or low level mouse/keyboard, etc. RL-GPT addresses this by trying to leverage the both, with the higher level actions appended to the RL agent’s action space.
- The state-of-the-art performance in MineDojo tasks highlights its potential impact on the field, though the focus on Minecraft limits the generalizability of the results.

**Weaknesses:**

1. The work focus on Minecraft, while useful as a testbed, may not fully represent the range of challenges present in other open-world environments. Expanding the scope of testing to other domains would enhance the significance of the work

2. While the integration approach is novel, the individual components (LLMs, RL, task decomposition) are well-known techniques. The paper could benefit from a more explicit discussion of how these components are synergistically combined to create a unique solution.

3. Some sections, particularly those describing the two-loop iteration process and the role of the critic agent, could benefit from more detailed explanations or examples to improve understanding. I am aware that the full prompts are included in the appendix and they do not fit in the main paper, however.

**Questions:**

1. The slow agent can all a sub-action that the RL agent executes. But can the fast agent which uses “code as policies” also call the RL agent inside its loop?
2. What is the architecture and inputs of the PPO RL agent? In particular, it was said that “PPO is limited to a set of skills”: how is the sub-action (e.g. “Harvest log” sub-action 2 in Figure 3) from the slow agent represented to the policy network? Is it just a one-hot vector (“PPO is constrained to a limited set of skills”), or are you encoding the sub-action text instruction as an embedding vector, etc., or are you learning a different set of policy weights for each sub-action, or something else?
3. How would the proposed framework adapt to other open-ended environments beyond Minecraft? Are there specific modifications required for different domains?
4. (RL Interface Ablation) Do you have any hypotheses on why the action space reconstruction is more effective than designing the reward function? Is the action space shortening the horizon of the task and makes the reward less sparse?
5. Table 4: is this showing the %? In Section 4.4, it was reported as 0.34% and 0.65% success rate. Was this a typo and it’s supposed to be 34% and 65% success percentage for crafting a table?

Post rebuttal: I have increased the score after the rebuttal.

**Limitations:**

The authors acknowledge some limitations of their work, notably the reliance on the capabilities of LLMs and the potential challenges in generalizing to unseen data. However, there are additional limitations and assumptions that could be more explicitly addressed:
1. The approach assumes an environment with a structured observation space that can support coding as policies. This might not be feasible in more complex or less structured environments. Environments need to be compatible with the two-level hierarchical framework and provide sufficient information for task decomposition and action coding. It’ll be helpful to provide guidelines for adapting the framework to different types of environments, including those with less structured observation spaces.
2. The framework's reliance on multiple interacting agents and iterative refinement could complicate implementation and debugging. The need for specific prompts and hand written examples add to the complexity, making it challenging for broader adoption.

---

> ### Author Rebuttal · Authors · 2024-08-07
>
> Dear Reviewer LWBp,
>
> Thank you for appreciating our work with valuable suggestions. We address your questions below.
>
> **Q1. Generalization to other environments.**
>
> **A1.** Thanks for this suggestion! We acknowledged this concern and addressed it to some extent in Appendix Section D. It is difficult to find real-world environments like Minecraft, which require **both high-level planning and low-level control**. We applied our methods to some robotic tasks that demand both long-horizon planning and precise motor execution. Some results are shown in **Fig. 7 and Fig. 8 in the attached pdf**.
>
> - Kitchen Environment Training: **Fig.7** illustrates the RL training process in the Kitchen environment[1]. The vertical axis represents the success rate, and the horizontal axis represents the number of training steps. Inserting coded motion planning into the action space accelerates learning. Our method learns faster compared to the baseline.
> - Furniture Environment Demonstration: In **Fig.8**, we present a qualitative demonstration of the Furniture environment[2]. The motion planning action effectively aids in hole-finding tasks during table and chair assembly. The baseline struggles to find the correct location at the same training step.
>
> [1] Relay Policy Learning: Solving Long-Horizon Tasks via Imitation and Reinforcement Learning, 2019
>
> [2] Furniture Assembly Environment for Long-Horizon Complex Manipulation Tasks, 2021
>
> Yes, modifications are needed for different domains, such as **adjusting the task descriptions in the prompts**. The powerful zero-shot capability of GPT should ensure generalization ability.
>
> **Q2. How are these components synergistically combined?**
>
> **A2.** Thanks for this suggestion! The key technical novelty is illustrated in **Fig.2** of the paper. Incorporating high-level actions generated by LLMs into the RL action space captures the core ideas presented. This core design allows the agent to choose between RL and code-as-policy.
>
> **Q3. More detailed explanations or examples.**
>
> **A3.** Thanks for this suggestion!
> - **Slow Agent:** This agent will decide which parts can be coded and which parts should be learned. The first iteration loop is designed to correct decision-making errors.
> - **Fast Agent:** This agent will code the coding tasks from the slow agent. The second iteration loop is designed to correct coding errors.
> - **Critic Agent:** The critic agent provides feedback from the environment for each code execution. The output from the critic agent after one step serves as feedback for the fast agent, while outputs from a sequence of steps serve as feedback for the slow agent.
>
> **Q4. Can the fast agent also call the RL agent inside its loop?**
>
> **A4.** Sorry for the confusion. The fast agent cannot perform RL training. Its role is to generate code based on the requirements provided by the slow agent.
>
> **Q5. The architecture and inputs of the PPO RL agent.**
>
> **A5.** Sorry for the confusion. Yes, we will learn the weights for each RL sub-action (marked orange in **Fig. 3**).
> 1. The "Harvest log" is sent from the slow agent to the fast agent in text format.
> 2. The fast agent generates high-level codes for the action.
> 3. Coded actions are inserted into the RL action space.
> 4. RL training is performed after this insertion.
>
> **Q6. Action space vs reward function.**
>
> **A6.** Thanks for this good question! We have conducted ablation studies, as shown in **Tab. 5**. Using an action space with higher-level actions can **shorten the task horizon and reduce reward sparsity**. This design is more efficient as it leverages the coding ability and common sense of LLMs. For instance, LLMs understand that it takes 10 attacks to break a tree. When designing a reward function, even though "10 attacks" yields a high reward, randomly sampling those 10 individual attacks is a time-consuming process. In contrast, with action space design, an action like "attack 10 times" can be directly generated, **resulting in an immediate high reward**.
>
> **Q7. Typos.**
>
> **A7.** Thank you for pointing that out. The correct values are 34% and 65%. We will correct this in the revision.

---

> > ### Comment · Reviewer_LWBp · 2024-08-14
> >
> > Thank you for the detailed response addressing my questions and concerns regarding generalization to other environments, how the components are combined and more detailed explanations. I have increased my score to 7.

---

> > > ### Author Response · Authors · 2024-08-14
> > >
> > > Thank you for your insightful comments and questions, which have greatly improved the quality of our paper. We deeply appreciate your recognition and the score increase! We will carefully integrate the results discussed with you into the revision.

---

### Author Rebuttal · Authors · 2024-08-07

Dear all reviewers,

We sincerely thank your effort in the review with valuable comments and suggestions. **We appreciate reviewers _LWBp_, _m13r_, and _efny_ for recognizing our work**. Additional figures are attached in the **_6379_rebuttal_figs.pdf_**, which we will reference in the following specific responses.

---

### Decision · Program_Chairs · 2024-09-25

**Decision:**

Accept (oral)

**Comment:**

This paper discusses how one can design an agent empowered by LLMs that can break down a task into subtasks and dynamically decide whether to solve such a subtask by code generated by the LLM or by a "traditional" RL agent. The method is validated in the game Minecraft. This is an interesting paper; all reviewers unanimously recommended its acceptance, and I'm doing the same.

However, an important point about the number of independent runs was made, and the concerns raised are real. I'm ok accepting this paper because it is a _proof of possibility. It shows how such an idea can be instantiated, and it shows that such an instantiation can work. This is how it should position itself instead of making claims that "it attains State-of-the-Art (SOTA) performance" (line 60). It is hard to imagine someone justifiably claiming SOTA performance without reporting some notion of variability or robustness. Thus, I strong suggest the authors to change their wording accordingly.

Additionally, some of the comparisons to RL methods are unfair, mainly when talking about their sample complexity. In line 75, for example, the paper states that the problem with other techniques is that they "requires billions of steps for fine-tuning long-horizon tasks". It is crucial to understand that LLMs are generated by going over many, many, many samples (orders of magnitude more than billions). Thus, the comparison above is unfair. LLMs allow one to amortize the cost of such samples because LLMs are a general-purpose model that can bypass the lack of initial knowledge of RL agents. I expect the final version of this paper to also better contrast the different alternatives as per this discussion.